# HD-cos Networks: Efficient Neural Architechtures for Secure Multi-Party Computation

## Abstract

Multi-party computation (MPC) is a branch of cryptography where multiple non-colluding parties execute a well designed protocol to securely compute a function. With the non-colluding party assumption, MPC has a cryptographic guarantee that the parties will not learn sensitive information from the computation process, making it an appealing framework for applications that involve privacy-sensitive user data. In this paper, we study training and inference of neural networks under the MPC setup. This is challenging because the elementary operations of neural networks such as the ReLU activation function and matrix-vector multiplications are very expensive to compute due to the added multi-party communication overhead. To address this, we propose the HD-cos network that uses 1) cosine as activation function, 2) the Hadamard-Diagonal transformation to replace the unstructured linear transformations. We show that both of the approaches enjoy strong theoretical motivations and efficient computation under the MPC setup. We demonstrate on multiple public datasets that HD-cos matches the quality of the more expensive baselines.

## 1 Introduction

Machine learning models are often trained with user data that may contain private information. For example, in healthcare patients diagnostics contain sensitive information and in financial sectors, user data contains potentially private information such as salaries and taxes. In these applications, storing the user data in plain text format at a centralized server can be privacy invasive. There have been several efforts to design secure and private ways of learning and inferring machine learning models. In this work, we focus on secure *multi-party computation* (MPC), a branch of cryptography that allows parties to collaboratively perform computations on data sets without revealing the data they possess to each other (Evans et al., 2017). Recently, there are several research papers that proposed to train and infer machine learning models in a secure fashion via MPC (Gilad-Bachrach et al., 2016; Graepel et al., 2012; Obla et al., 2020). Loosely speaking, in the MPC setup, a piece of sensitive data is split into multiple shards called secret shares, and each secret share is stored with a different party. These parties are further chosen such that their fundamental interests are to protect sensitive data and hence can be viewed as non-colluding. The cryptography guarantee states that unless all the parties (or at least k out of all parties, depending on the cryptographic algorithm design) collude, sensitive data cannot be reconstructed and revealed to anyone/anything.

Training in the MPC setup is challenging due to several reasons. Firstly, only limited data types (e.g. integer) and/or operations (e.g. integer addition and multiplication) are natively supported in most MPC algorithms. This increases the complexity to support non-trivial operations over MPC. Secondly, since data is encrypted into multiple secret shares and stored in multiple parties, one share per party, directly training a machine learning model on this data can be expensive, both in terms of communication between the servers and computation at the individual servers. More concretely if simple operations like addition and multiplications take orders of nanoseconds in the normal computation scenarios, in the MPC setup they can take milliseconds or more, if the operation requires the parties in the MPC setup to communicate with each other. Furthermore, one of the key bottlenecks of secret sharing mechanisms is that most non-linear operations e.g., $\text{ReLU}(x) = \max(0, x)$, cannot be efficiently computed.

In this paper, we address these questions by proposing a general network construct that can be implemented in MPC setup efficiently. Our proposal consists of two parts: 1) use the cosine function as the activation function, and 2) use a structured weight matrix based on the Hadamard transform in place of the standard fully connected layer. We provide an algorithm to compute cosine under two-party computation (2PC) setup. Unlike ReLU which involves multiple rounds of communication, our proposed algorithm for cosine requires only two online rounds of communication between the two computation servers. The use of the proposed Hadamard transform for weight matrices means that the number of parameters in each dense layer scales linearly with the input dimenion, as opposed to the standard dense layer which scales quadratically. We demonstrate on a number of challenging datasets that the combination of these two constructs leads to a model that is as accurate as the commonly used ReLU-based model with fully connected layers.

The rest of the paper is organized as follows. We first overview multiparty computation in Section 2 and then overview related works in Section 3. We present the cosine activation function and structured matrix transformation with theoretical motivations and analysis of their computational efficiency in Section 4 and Section 5. We then provide extensive experimental evaluations on several datasets in Section 6.

## 2 MULTI-PARTY COMPUTATION

*Secure multi-party computation* (MPC) is a branch of cryptography that allows two or more parties to collaboratively perform computations on data sets without revealing the data they possess to each other (Evans et al., 2017). Following earlier works (Liu et al., 2017; Mohassel & Zhang, 2017; Kelkar et al., 2021), we focus on the two party computation (2PC) setup. Our results can be extended to multi-party setup by existing mechanisms. If the two parties are non-colluding, then 2PC setup guarantees that the two parties will not learn anything from the computation process and hence there is no data leak.

During training in the 2PC setup, each party receives features and labels of the training dataset in the form of secret shares. They compute and temporarily store all intermediate results in the form of secret shares. Thus during training, both the servers collaboratively learn a machine learning model, which again is split between two parties. Upon completion of the training process, the final result, i.e. the ML model itself, composed of trained parameters, are in secret shares to be held by each party in the 2PC setup. At prediction time, each party receives features in secret shares, and performs the prediction where all intermediate results are in secret shares. The MPC cluster sends the prediction results in secret share back to the caller who provided the features for prediction. The caller can combine all shares of the secret prediction result into its plaintext representation. In this entire training/prediction process, the MPC cluster does not learn any sensitive data.

While MPC may provide security/privacy guarantee and is Turing complete, it might be significantly slower than equivalent plaintext operation. The overall performance of MPC is determined by the computation and communication cost. The computation cost in MPC is typically higher than the cost of equivalent operations in cleartext. The bigger bottleneck is the communication cost among the parties in the MPC setup. The communication cost has three components.

- **Number of rounds**: the number of times that parties in the MPC setup need to communicate/synchronize with each other to complete the MPC crypto protocol. For 2PC, this is often equivalent to the number of Remote Procedure Calls (or RPCs) that the two parties need per the crypto protocol design. Many MPC algorithms differentiate offline rounds vs. online rounds, where the former is input-independent and can be performed asynchronously in advance, and the latter is input-dependent, is on the critical path for the computation and must be performed synchronously. For example, addition of additive secret shares requires no online rounds, whereas multiplication of additive secret shares requires one online round.

- **Network bandwidth**: the number of bytes that parties in the MPC setup need to send to each other to complete the MPC crypto protocol. For 2PC, each RPC between the two parties has a request and response. The bandwidth cost is the sum of all the bytes to be transmitted in the request and response per the crypto protocol.

- **Network latency**: the network latency is a property of the network connecting all parties in the MPC setup. It depends on the network technology (e.g. 10 Gigabit Ethernet or 10GE), network topology, as well as applicable network Quality of Service (QoS) settings and the network load.

In this work, we propose neural networks which can be implemented with a few online rounds of communication and little network bandwidth. Following earlier works , We consider neural network architectures where the majority of the parameters are on the fully connected layers.

- We propose and systematically study using cosine as the activation and demonstrate that it achieves comparable performance to existing activation and can be efficiently implemented with two online rounds.
- We show that dense matrices in neural networks can be replaced by structured matrices, which have comparable performance to existing neural network architectures and can be efficiently implemented in MPC setup by reducing the bandwidth. We show that by using structured matrices, we can reduce the number of per-layer secure multiplications from $\mathcal{O}(d^2)$ to $\mathcal{O}(d)$, where $d$ is the layer width, thus reducing the memory bandwidth.

## 3 RELATED WORKS

**Neural network inference under MPC.** Barni et al. (2006) considered inference of neural networks in the MPC setup, where the linear computations are done at the servers in the encrypted field and the non-linear activations are computed at the clients directly in plaintext. The main caveat of this approach is that, to compute a $L$ layer neural network, $L$ rounds of communication is required between server and clients which can be prohibitive and furthermore intermediate results are leaked to the clients, which may not be desired. To overcome the information leakage, Orlandi et al. (2007) proposed methods to hide the results of the intermediate data from the clients; the method still requires multiple rounds of communication. Liu et al. (2017) proposed algorithms that allows for evaluating arbitrary neural networks; the intermediate computations (e.g., $\text{sign}(x)$) are much more expensive than summations and multiplications.

**Efficient activation functions.** Since inferring arbitrary neural networks can be inefficient, several papers have proposed different activation functions which are easy to compute. Gilad-Bachrach et al. (2016) proposed to use the simple square activation function. They also proposed to use mean-pooling instead of max-pooling. Chabanne et al. (2017) also noticed the limited accuracy guarantees of the square function and proposed to approximate ReLU with a low degree polynomial and added a batch normalization layer to improve accuracy. Wu et al. (2018) proposed to train a polynomial as activation to improve the performance. Obla et al. (2020) proposed a different algorithm for approximating activation methods and showed that it achieves superior performance on several image recognition datasets. Recently Knott et al. (2021) released a library for training and inference of machine learning models in the multi-party setup.

**Cosine as activation function.** Related to using cosine as the activation function, Yu et al. (2015) proposed the Compact Nonlinear Maps method which is equivalent of using cosine as the activation function for neural networks with one hidden layer. Parascandolo et al. (2016) studied using the sine function as activation. Xie et al. (2019) used cosine activation in deep kernel learning. Noel et al. (2021) proposed a variant of the cosine function called Growing Cosine Unit $x\cos(x)$ and shows that it can speed up training and reduce parameters in convolutions neural networks. All the above works are not under the MPC setup.

**Training machine learning models under MPC.** Graepel et al. (2012) proposed to use training algorithms that can be expressed as low degree polynomials, so that the training phase can be done over encrypted data. Aslett et al. (2015) proposed algorithms to train models such as random forests over training data. Mohassel & Zhang (2017) proposed a 2PC setup for training and inference various types of models including neural networks where data is distributed to two non-colluding servers. Kelkar et al. (2021) proposed an efficient algorithm for Poisson regression by adding a secure exponentiation primitive.

**Combining with differential privacy.** We note that while the focus of this work is multi-party computation, it can be combined with other privacy preserving techniques such as differential privacy

| Activation | Online rounds |
|---|---|
| None | 0 |
| $x^2$ (Gilad-Bachrach et al., 2016) | 1 |
| $e^x$ - 1 (Kelkar et al., 2021) | 1 |
| ReLU Polyfit(3) (Obla et al., 2020) | 2 |
| **Cosine** [this work] | 2 |

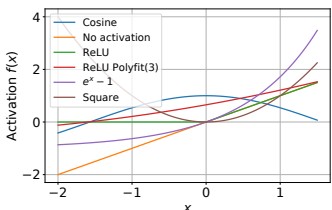

Table 1: Activation functions and their communication costs. Since ReLU cannot be implemented efficiently in the MPC setup, we have omitted it.

Figure 1: A comparison of different activation functions.

(Dwork et al., 2014). There are some recent works which combine MPC with differential privacy (Jayaraman & Wang, 2018). Systematically evaluating performances with the combination of our proposed technique and that of differential privacy remains an interesting future direction.

## 4 COSINE AS ACTIVATION

The most widely used activation function in the non-MPC setup is the Rectified Linear Unit (ReLU) function: $\text{ReLU}(x) = \max(0, x)$. Unfortunately, it is not efficient to compute ReLU under the MPC setup. To this end, Wu et al. (2018) and Obla et al. (2020) proposed to approximate ReLU with low-degree polynomials. This still incurs large communication cost as it requires $d - 1$ online online rounds of communications to compute $d$-degree polynomial approximation. Another simple alternative is the square function (Gilad-Bachrach et al., 2016). In this case, the trained neural network often has a subpar quality as noted by Wu et al. (2018).

In this work, we propose to use cosine as the activation function. We show that it has good theoretical properties (Section 4.1) and can be implemented efficiently under the 2PC setup (Section 4.2). We compare different activation functions in Figure 1 and give the corresponding computation costs in Table 4.

### 4.1 THEORETICAL MOTIVATION OF COSINE

When using cosine as the activation function, the output of a neural network layer becomes $\cos(Wx + b)$, where $x \in \mathbb{R}^d$ is the input, $W \in \mathbb{R}^{k \times d}$ is the weight matrix, and $b \in \mathbb{R}^k$ is a bias vector. This form coincides with the Random Fourier Feature method (Rahimi & Recht, 2007), widely used in the kernel approximation literature. Kernel method is a type of powerful nonlinear machine learning models that is computationally expensive to scale to large-scale datasets. Kernel approximation methods aim at mapping the input into a new feature space, such that the dot product in that space approximate the value of the kernel. The benefit is that a linear classifier trained in the mapped space approximate a kernel classifier.

Let $\phi(x) = \cos(Wx + b)$, the follow result in Rahimi & Recht (2007) shows that $\phi(x) \cdot \phi(y)$ approximates a shift-invariant kernel with proper choice of the parameters in $W$ and $b$. Note that since (Rahimi & Recht, 2007), there have been many improvements of Random Fourier Features such as better rate of approximation (Sriperumbudur & Szabó, 2015) and more efficient sampling (Yang et al., 2014). Cosine is also shown to be able to approximate other kernel types such as the polynomial kernels on unit sphere (Pennington et al., 2015).

**Lemma 1** (Cosine approximates a shift-invariant kernel (Rahimi & Recht, 2007)). *Let $k \colon \mathbb{R}^d \times \mathbb{R}^d \to \mathbb{R}$ be a positive definite kernel. Assume that $k$ is shift-invariant: there exists a function $K \colon \mathbb{R}^d \to \mathbb{R}$ such that for any $x, y \in \mathbb{R}^d$, $K(x - y) = k(x, y)$. Let $\phi(x) = \sqrt{2/D}[\cos(w_1 \cdot x + b_1), ..., \cos(w_D \cdot x + b_D)]$, where $w_1, \ldots, w_D$ are sampled i.i.d. from a distribution $p(w)$ which is the Fourier transformation of $K(z)$ i.e., $K(z) = \int_{\mathbb{R}^d} p(w) e^{jw \cdot z} dw$. $b_1, ..., b_D$ are sampled uniformly from $[0, 2\pi]$. Then $\phi(x) \cdot \phi(y)$ is an unbiased estimator of $K(x - y)$, and*

$$\mathbb{P}\left[\sup_{x,y \in \mathcal{M}} |\phi(x) \cdot \phi(y) - k(x - y)| \geq \epsilon\right] \leq C \left(\frac{diam(\mathcal{M})}{\epsilon}\right)^2 e^{-D\epsilon^2/d},$$

*where $\mathcal{M}$ is a compact subset of $\mathbb{R}^d$, $diam(\mathcal{M})$ is the diameter of $\mathcal{M}$, and $C$ is a constant.*

---

**Input:** server$_1$ has $[x]_1$, and server$_2$ has $[x]_2$ such that $[x]_1 + [x]_2 = x$.
**Output:** server$_1$ has $[z]_1$, and server$_2$ has $[z]_2$ such that $[z]_1 + [z]_2 = \cos(x)$.

1. **Local computation:** Both servers compute $\cos[x]_i$ and $\sin[x]_i$ locally.
2. **Exchange:**
   - From $\cos[x]_1$, server$_1$ constructs $[\cos[x]_1]_1$ and $[\cos[x]_1]_2$ using additive secret share mechanisms (Evans et al., 2017); sends $[\cos[x]_1]_2$ to server$_2$.
   - From $\sin[x]_1$, server$_1$ constructs $[\sin[x]_1]_1$ and $[\sin[x]_1]_2$; sends $[\sin[x]_1]_2$ to server$_2$.
   - From $\cos[x]_2$, server$_2$ constructs $[\cos[x]_2]_1$ and $[\cos[x]_2]_2$; sends $[\cos[x]_2]_1$ to server$_1$.
   - From $\sin[x]_2$, server$_2$ constructs $[\sin[x]_2]_1$ and $[\sin[x]_2]_2$; sends $[\sin[x]_2]_1$ to server$_1$.
3. **Multiplication:**

   $[\cos[x]_1\cos[x]_2]_1, [\cos[x]_1\cos[x]_2]_2 = \text{Mult}(([\cos[x]_1]_1, [\cos[x]_2]_1), ([\cos[x]_1]_2, [\cos[x]_2]_2)).$

   $[\sin[x]_1\sin[x]_2]_1, [\sin[x]_1\sin[x]_2]_2 = \text{Mult}(([\sin[x]_1]_1, [\sin[x]_2]_1), ([\sin[x]_1]_2, [\sin[x]_2]_2)).$

4. **Final computation:**
   server$_1$ computes $[z]_1 = [\cos[x]_1\cos[x]_2]_1 - [\sin[x]_1\sin[x]_2]_1$.
   server$_2$ computes $[z]_2 = [\cos[x]_1\cos[x]_2]_2 - [\sin[x]_1\sin[x]_2]_2$.

---

**Algorithm 1:** Securely compute cosine activation in the 2PC setup. Mult() refers to the known secure multiplication protocol based on Beaver Triplets (Beaver, 1991).

The above lemma shows that, at random initialization, a linear classification model trained with the transformed features using cosine approximates a kernel-based classifier. In specific, if the weights $\{w_i\}_{i=1}^D$ are sampled from a Gaussian distribution, the model is approximating a Gaussian-kernel based classifier, already a strong baseline at initialization.

Besides the connection the kernel methods, cosine and its derivative are bounded functions. As shall be seen in our experiments, having a linearly-growing (e.g., ReLU) or a bounded activation (e.g., cos) means that the model can be trained without carefully fine-tuning the optimization method. That is, models with the cosine activation can be trained with a wide range of learning rates, in contrast to, for instance, the square function which grows quickly enough that a careful choice of learning rate is needed to prevent numerical overflow. Note that techniques like batchnorm that improves stability in neural network training are hard to be applied under the MPC setup due to the expensive operation of division. Hence, having a numerically stable activation function is very important.

## 4.2 ALGORITHM TO COMPUTE COSINE ACTIVATION FUNCTION

In this section, we provide an algorithm to compute cosine activation function in the 2PC setup. The algorithm is given in Figure 1. We state the algorithm in the bracket notation, where $[y]_i$ denote the shard of $y$ located in server$_i$. Therefore $[y]_1 + [y]_2 = y$. The algorithm has two main parts: locally the algorithms first compute $\cos[x]_i$ and $\sin[x]_i$. Then they use known 2PC protocols for addition, multiplication and secure exchange to compute $[z]_1$ and $[z]_2$ such that $[z]_1 + [z]_2 = z$ and

$$z = \cos[x]_1 \cos[x]_2 - \sin[x]_1 \sin[x]_2.$$

By the trignometric identity $\cos(a+b) = \cos(a)\cos(b) - \sin(a)\sin(b)$,

$$z = \cos([x]_1 + [x]_2) = \cos(x).$$

**Theorem 1.** *Cosine can be computed in the 2PC setup with two online rounds of communication.*

*Proof.* Observe that in Algorithm 1 the overall computation only requires known 2PC protocols for addition, secure exchange, and multiplication. Addition does not require any online rounds. It is known that multiplication can be implemented using Beaver Triplets Beaver (1991) in one online round. Finally secure exchange requires one online round of communication. Hence the total number of online rounds of communication is two. □

## 5 Fast linear transformation using the Hadamard-Diagonal matrices

One elementary computation in neural network is linear transformation: $y = Wx$, where $x \in \mathbb{R}^d, W \in \mathbb{R}^{k \times d}, y \in \mathbb{R}^k$. This operation is expensive due to the $\mathcal{O}(dk)$ multiplications involved. In the MPC setup, both $W$ and $x$ are encrypted and thus computing their product requires bandwidth proportional to $\mathcal{O}(dk)$, which can be prohibitive.

In this paper we propose to impose specific types of structure on the $W$ matrix. For simplicity, let us assume $k = d$.[1] We propose using $W = HD$, where $D \in \mathbb{R}^{d \times d}$ is a diagonal matrices with learnable weights, and $H$ is the normalized Walsh-Hadamard matrix of order $d$ with the following recursive definition:

$$H_1 = [1], \quad H_2 = \frac{1}{\sqrt{2}} \begin{bmatrix} 1 & 1 \\ 1 & -1 \end{bmatrix}, \quad H_{2^k} = \frac{1}{2^{k/2}} \begin{bmatrix} H_{2^{k-1}} & H_{2^{k-1}} \\ H_{2^{k-1}} & -H_{2^{k-1}} \end{bmatrix}.$$

### 5.1 Theoretical motivations

Speeding up linear transformations $y = Wx$ has been an extensively studied topic under different applications. For example, in dimensionality reduction ($k < d$), when the element of $W$ are sampled iid from a Gaussian distribution, the well-known Johnson-Lindenstrauss lemma states that the l2 distance of the original space is approximately preserved in the new space. Here $W$ can be replaced by structured matrices such as the fast Johnson-Lindenstrauss transformation (a sparse matrix, a Walsh-Hadamard matrix, and a random binary matrix) (Ailon & Chazelle, 2006), sparse matrices (Matoušek, 2008) and circulant matrices (Hinrichs & Vybíral, 2011). Such matrices give faster computation with similar error bound. Another example is binary embedding $y = \text{sign}(Wx)$. When elements of $W$ are sampled iid from the standard Gaussian distribution, the Hamming distance of the new space approximates the angle of the original space. Similarly, this operation can be made faster with structured matrices such as circulant (Yu et al., 2018) and bilinear matrices (Gong et al., 2013). Several types of structures have also been applied in speeding up the fully connected layers in neural networks. Examples are circulant (Cheng et al., 2015), fastfood (Yang et al., 2015), low-displacement rank (Thomas et al., 2018) matrices.

The reason why we chose to use the Hadamard-Diagonal structure goes back to the literature of kernel approximation. It was shown that by using such a structure, the mapping $y = \cos(Wx)$ provides even lower approximation error of a shift-invariant kernel (Yu et al., 2016) in comparison with the random unstructured matrices. Notice that other types of structures such as "fastfood" (Le et al., 2013) and circulant do not have such good properties. By pairing cosine with the Hadamard-Diagonal structure, even at initialization, the feature space already well mimic that of a kernel. Furthermore, by optimizing the weights of the diagonal matrices, we are implicitly learning a good shift invariant kernel for the task. In Section 6 we show that this structure almost does not hurt the quality of the model while reducing the computational cost. We also show in the appendix that cosine is indeed performing better than circulate and fastfood.

### 5.2 Cost of computing the Hadamard-Diagonal transformation

There are two operations in computing the Hadamard-Diagonal transformation.

1. Multiplication with a variable diagonal matrix. This involves $\mathcal{O}(d)$ secure multiplications.

2. Multiplication with a fixed Hadamard matrix. Since the Hadamard matrix is a linear transformation that is fixed and publicly known, multiplying a vector with Hadamard matrix requires $\mathcal{O}(d \log d)$ sums and subtractions on each server locally and does not require any communication between the servers.

Thus computing $HDx$ for a vector $x$, requires $\mathcal{O}(d \log d)$ sums and subtractions locally and $\mathcal{O}(d)$ bits of communication. This is significantly smaller than the traditional matrix vector multiplication, which requires $\mathcal{O}(d^2)$ bits of communication between the two servers. Note that one can simply

---

[1]$k < d$ can be handled with picking the first $k$ elements of the output. $x$ can be padded with 0s such that $d$ is a power of 2.

improve the model capacity by using multiple $HD$ blocks such as $y = HD_3HD_2HD_1x$. We empirically observe that more blocks only provides marginal quality improvements, yet much high computational cost. So in this work, we advocate using a single $HD$ block.

# 6 EXPERIMENTS

In this section, we study how modeling choices affect the model prediction accuracy. These choices include the optimization method, activation function, number of layers, and structure of weight matrices. For all the experiments, we use SGD as the optimizer as it is known that implementing optimizers which compute moments such as Adam incurs significantly more overhead in MPC setup. For all the datasets, we sweep learning rates in the set {1e-5, 1e-4, 1e-3, 0.01, 0.1, 0.5}, and show the best averaged result over five trials. We note that our goal here is not to produce the state-of-the-art results on datasets we consider. In fact under the MPC setup, it is not practical to use many more advanced architectures (e.g. attention), optimizes (e.g. Adam) and techniques (e.g. batchnorm). Rather, we aim to investigate the predictive performance of simple models that are suitable for the MPC setup. More specifically, we provide empirical evidence for the following important findings:

1. Cosine activation performs better than other existing activation functions in the 2PC setup. Furthermore, the performance is significantly superior if the network has many layers.

2. Replacing dense matrices with Hadamard-diagonal matrices incurs only a small performance degradation.

We report additional results in the appendix. There, we show that cosine activation is more stable to train compared to square activation and hence is preferred in the MPC setup. While the use of an adaptive optimization procedure like Adam (Kingma & Ba, 2014) is limited in the MPC setting, for completeness, we report our results with Adam in the appendix, where we show that cosine still performs better than other existing activation functions. In the main text, we will focus on SGD as our optimization method of choice. For all experimental results reported, the standard deviation computed across trials is in the order of 0.1 or smaller i.e., the only factor of variability is the initialization. We omit the standard deviation for brevity.

## 6.1 DATASETS AND MODELS

**Datasets** We consider four classification datasets: MNIST, Fashion-MNIST, Higgs, and Criteo. These data cover challenging problems from computer vision, physics and online advertising domains, respectively. Briefly the MNIST and Fashion-MNIST datasets are 10-class classification problems where each example is an image of size $28 \times 28$ pixels. The Higgs dataset[2] represents a problem of detecting the signatures of Higgs Boson from background noise (Baldi et al., 2014). This is a binary classification problem where each record consists of 28 floating-point features representing kinematic properties. We randomly subsample 7.7M records (70%) from the available 11M examples, and use them for training. The remaining 3.3M records are used for testing. The Criteo dataset is a challenging ad click-through rate prediction problem, and was used as part of a public Kaggle competition.[3] There are 36 million records in total with 25.6% labeled as clicked. We preprocess the Criteo data following the setup of Wang et al. (2017). More technical details regarding these datasets and propcessing steps can be found in Section A in the appendix.

**Models** For MNIST and Fashion-MNIST, we use LeNet-5 (LeCun et al., 1998) as the model, where the activation functions in the last two hidden layers are replaced. For Higgs data, we use a Multi-Layer Perceptron (MLP) model with four hidden layers, each with 16 units. A discussion on the number of layers and more results are presented in the appendix to show training stability of different activation functions. For Criteo, we follow Wang et al. (2017) and use an MLP with four hidden layers, with 1024 hidden units in each. In all cases, we consider three variants for each dense layer in the model: 1) the standard dense (fully connected) layer, 2) the fast Hadamard-Diagonal layer as

---

[2]Higgs dataset is available at `https://www.tensorflow.org/datasets/catalog/higgs`.

[3]The Criteo dataset was used as a prediction problem as part of the Display Advertising Challenge in 2014 (a Kaggle competition) by Criteo (`https://www.kaggle.com/c/criteo-display-ad-challenge`).

| Activation | MNIST | Fashion-MNIST | Higgs | Criteo |
|---|---|---|---|---|
| Cosine | **99.1** | **88.0** | **74.4** | 57.8 |
| $e^x - 1$ | 98.0 | 86.6 | 50.0 | 57.1 |
| None | 92.2 | 83.6 | 63.5 | 57.0 |
| ReLU Polyfit(3) | 99.0 | 87.4 | 64.8 | 57.5 |
| Square | 80.9 | 61.8 | 50.0 | **59.0** |
| ReLU (non MPC baseline) | 99.1 | 90.0 | 74.2 | 59.5 |

Table 2: Summary of results on all the datasets using the standard unstructured dense layer. We report test accuracy except Criteo where we report AUC-PR due to its skewed label distribution. Even though ReLU cannot be computed in the MPC setup, we state its performance for reference.

proposed in Section 5, and 3) a dense layer where the weight matrix has rank at most two i.e., the weight matrix $W = V_1^\top V_2$ where both matrices $V_1$ and $V_2$ have two rows.

## 6.2 ACTIVATION FUNCTIONS

Table 2 summarizes our results for different activation functions. Recall that for each activation function, we report the best trial-averaged performance across a number of learning rate candidates (trained with SGD). Here we use the standard dense layer, as opposed to a structured weight matrix. We report the test accuracy for all datasets, except for Criteo where we instead report AUC-PR which is more appropriate for its skewed label distribution. We observe that our proposed cosine activation function performs as well as the standard activation function like ReLU, while being more computationally tractable for the 2PC setup. To understand other activation functions, consider the example of Higgs dataset. In this dataset, using no activation function yields an accuracy of 63.5%, whereas $e^x - 1$ and Square only performance at the chance level since theses models struggle to train without a numerical issue. Specifically, a closer inspection reveals that these activation functions grow fast, and require a careful choice of the optimization

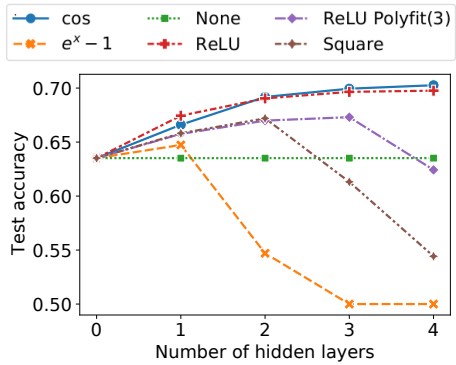

Figure 2: Test accuracy as a function of the number of hidden layers in the Higgs problem. We use the proposed Hadamard-Diagonal layer with each activation function listed.

method, and learning rate to prevent numerical overflow. By contrast, cosine is bounded and is more robust to a sub-optimal choice of the optimization algorithm (more on this point in the appendix).

We also investigated the performance of these activations as a function of number of layers for the Higgs dataset. The results are given in Figure 2 where the proposed Hadamard structure is used for all activation functions. While most activation functions perform well with fewer hidden layers, only cosine and ReLU performs well as the number of layers increases. This is because these functions grow at most linearly as their derivatives are bounded by one, and do not require a careful tuning of the learning rate to prevent numerical overflow. We note that the same observation holds true when we use the standard dense layer, and a low rank weight matrix. We omit these results for brevity. More discussion on this aspect can be found in Section B.2 in the appendix.

## 6.3 STRUCTURED MATRICES

Next we investigate the effect of using a structured weight matrix in each dense layer. To this end, we consider the cosine activation function for all datasets, and consider replacing each dense layer with one of the three variants described in Section 6.1: 1) the standard dense layer (Dense), 2) the proposed Hadamard-Diagonal layer (Hadamard) as described in Section 5, and 3) dense layer with a weight matrix of at most rank two (Low rank). We present test accuracy and AUC-PR in Table 3. It can be seen that using the Hadamard weight matrix incurs a minor loss of performance across different datasets, compared to the standard dense layer. Recall that the number of parameters in

the proposed Hadamard-Diagonal layer is only linear in the number of inputs, as opposed to being quadratic as in the standard dense layer. Interestingly the performance of the low-rank weight matrix is lower than that of the Hadamard weight matrix, even though it has four times as many learnable parameters. This hints that having an orthogonal weight matrix helps increase expressiveness of the model, presumably because it forms a basis in the latent feature space of the network.

| Structure | # secure mult. | MNIST | Fashion-MNIST | Higgs | Criteo |
|-----------|----------------|-------|---------------|-------|--------|
| Dense | $d^2$ | 99.1 | 88.0 | 74.4 | 57.8 |
| Low rank | $4d$ | 94.4 | 84.9 | 67.9 | 25.6 |
| Hadamard | $d$ | 98.5 | 88.9 | 70.3 | 58.0 |

Table 3: Test accuracy on the four problems with cosine activation function. For Criteo where the label distribution is skewed, we report AUC-PR.

In fact, we observe that the superiority of the Hadamard-Diagonal layer to the low-rank weight matrix is not specific to cosine. We provide evidence in Table 4 where we report performance on Fashion-MNIST and Criteo for each combination of activation function and type of weight structure. Note that for Criteo, since the two classes are skewed with roughly 25% of the data belonging to the positive class, we report AUC-PR instead of the test accuracy. We observe that in most cases our proposed Hadamard structure yields models with higher performance than the low rank approach does for all activation functions, while being competitive to the standard dense layer (which has an order of magnitude more parameters). We include comparisons to more baselines in Table 9 in the Appendix: PHD (Yang et al., 2015) and Circulant matrix (Cheng et al., 2015). There too we find Hadamard tranform to be the best performing approach.

| | Fashion-MNIST | | | Criteo | | |
|---|---|---|---|---|---|---|
| Activation | Dense | Low rank | Hadamard | Dense | Low rank | Hadamard |
| Cosine | 88.0 | 84.9 | **88.9** | 57.8 | 25.6 | **58.0** |
| $e^x - 1$ | **86.6** | 54.2 | 85.9 | 57.1 | 57.4 | 57.4 |
| None | **83.6** | 73.8 | **83.6** | 57.0 | 56.8 | 48.5 |
| ReLU Polyfit(3) | 87.4 | 83.8 | **88.7** | 57.5 | 25.6 | 52.7 |
| Square | **61.8** | 35.0 | 35.7 | **59.0** | 25.6 | 25.6 |
| ReLU (non MPC baseline) | **90.0** | 86.3 | 89.2 | 59.5 | 57.7 | 46.6 |

Table 4: Test accuracy of the LeNet-5 model (LeCun et al., 1998) on Fashion-MNIST for each combination of activation function and type of the weight matrix. For Criteo, we use a standard feed-forward MLP with four hidden layers, each containing 1024 hidden units. Our proposed Hadamard structure works better than the low rank approach in most cases, for all activation functions, while being competitive to the standard dense layer (where no structure is imposed).

# 7    DISCUSSION AND FUTURE WORK

The HD-cos approach offers a generic, computationally efficient building block for training and inference of a neural network in the multi-party computation setup. The efficiency gain is a result of two novel ideas: 1) use a fast Hadamard transform in place of the standard dense layer to reduce the number of parameters from $d^2$ to $d$ parameters where $d$ is the input dimension; 2) use cosine as the activation function. As we demonstrated, the smoothness and boundedness of its derivative helps ensure robust training even when the learning rate is not precisely chosen. Both of these properties are of obvious value; yet a natural question remains: can we reduce the computational cost even further? A potential idea is to combine HD-cos with Johnson–Lindenstrauss random projection, which reduces the input dimensionality (without learnable parameters) while preserving distances. Investigating this would be an interesting topic for future research.

## 8 REPRODUCIBILITY STATEMENT

For all experiments, unless stated otherwise, the hyperparameters are set to the default values as provided by Tensorflow. Instructions to download datasets used in this work and processing steps necessary to reproduce our results are described in Section 6.1. The proposed HD-cos approach consists of known building blocks (i.e., cosine, and the fast Hadamard transform) and are thus easy to apply. Our contributions are from showing their utility in the Multi-Party Computation (MPC) setup. To our knowledge, no previous work has attempted to study cosine and the fast Hadamard transform in the context of MPC.

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

# HD-cos Networks: Efficient Neural Architechtures for Secure Multi-Party Computation

## Supplementary Material

## A Datasets and Preprocessing

This section contains technical details of all datasets and preprocessing we use in experiments in the main text.

**MNIST and Fashion-MNIST** The MNIST database contains 60,000 training and 10,000 test images of handwritten digits (LeCun et al., 2010). Each image is of size $28 \times 28$ pixels and represents one of the digits (0 to 9). The Fashion-MNIST dataset (Xiao et al., 2017) is a drop-in replacement for MNIST with the same image size and dataset size. Here each example is an image of one of the ten selected fashion products e.g., T-shirt, shirt, bag. For these two datasets, models are trained for 200 epochs with a batch size of 128.

**Higgs** The Higgs dataset represents a problem of detecting the signatures of Higgs Boson from background noise (Baldi et al., 2014). This is a binary classification problem where each record consists of 28 floating-point features, of which 21 features representing kinematic properties, and additional seven features are derived from the 21 features by physicists to help distinguish the two classes. The dataset is regarded as a challenging multivariate problem for benchmarking nonparametric two-sample tests i.e., statistical tests to detect the difference of between the distributions of the two classes (Chwialkowski et al., 2015; Liu et al., 2020). Out of the available 11M records, we randomly subsample 7.7M records (70%) and use them for training. No feature normalization or standardization is performed. The remaining 3.3M records are used for testing. We train models on Higgs data for 40 epochs with a batch size of 256.

**Criteo** The Criteo dataset is a challenging ad click-through rate prediction problem, and was used as part of a public Kaggle competition.[4] Here, the label is binary (clicked or not clicked), and the feature vector encodes user information and the page being viewed. Each feature vector consists of 39 features, of which 13 are integer and 26 are categorical. There are 36 million records in total with 25.6% labeled as clicked. Following Wang et al. (2017), we split the data into 80%/10%/10% for training, evaluation and testing, respectively. For each categorical feature, each value in its range is mapped to a trainable token embedding vector whose length is given by $6 \times$ (cardinality of the feature)$^{1/4}$. Each integer feature $x \in \mathbb{N}$ is mapped to $\log(1 + x)$. Concatenating all the preprocessed features gives a dense vector of length 1023, from which we feed to a standard multi-layer perceptron. All the models are trained for 150k update steps with a batch size of 512.

---

[4]The Criteo dataset was used as a prediction problem as part of the Display Advertising Challenge in 2014 (a Kaggle competition) by Criteo (https://www.kaggle.com/c/criteo-display-ad-challenge).

## B  ADAPTIVE OPTIMIZATION AND TRAINING STABILITY

In the main text, we omit the results for all learning rates and choose to present only the best performance across learning rates for each activation and type of weight structure, for the sake of brevity. In this section, we report results for all learning rates, study the interplay between the network depth and the choice of the activation function, as well as investigate the use of an adaptive optimization method like Adam (Kingma & Ba, 2014).

Note again that for all experimental results reported in this paper, the standard deviation computed across trials is in the order of 0.1 or smaller i.e., the only factor of variability is the initialization. We omit the standard deviation for brevity.

### B.1  OPTIMIZATION ALGORITHMS

We show that the optimization method used affects the performance of each activation function differently. To demonstrate, we consider the MNIST dataset and use the LeNet-5 (LeCun et al., 1998) model; this is a convolutional neural network with two convolution layers interleaved with average pooling layers, followed by two dense layers. Each of the convolution and dense layers has an activation function. Activation function candidates are cosine (proposed), exponential, none (no activation function), ReLU, ReLU Poly(3), and square. The ReLU Poly(3) is a degree-3 polynomial approximation of ReLU investigated in (Obla et al., 2020). Unlike ReLU which is expensive to implement in the MPC setup (i.e., requires scalar comparison), a polynomial only relies on multiplication and addition, and is more suitable for MPC. We consider $f(x) = e^x - 1$ instead of $e^x$ so that $f(0) = 0$, a property common to many activation functions. Table 5 shows the test accuracy for different optimization methods: Stochastic Gradient Descent (SGD) vs Adam (Kingma & Ba, 2014), learning rates, and activation functions.

| Optimizer | Learning rate Activation | $1e^{-5}$ | $1e^{-4}$ | $1e^{-3}$ | 0.01 | 0.1 | 0.5 |
|---|---|---|---|---|---|---|---|
| Adam | Cosine | .96 | .99 | .92 | .98 | .54 | .10 |
| | $e^x - 1$ | .97 | .98 | .99 | .10 | .10 | .10 |
| | None | .91 | .92 | .92 | .91 | .91 | .91 |
| | ReLU | .97 | .99 | .98 | .98 | .45 | .10 |
| | ReLU Poly(3) | .90 | .98 | .98 | .86 | .10 | .10 |
| | Square | .96 | .99 | .69 | .81 | .10 | .10 |
| SGD | Cosine | .69 | .95 | .97 | .98 | .92 | .10 |
| | $e^x - 1$ | .93 | .63 | .39 | .10 | .10 | .10 |
| | None | .88 | .91 | .91 | .92 | .91 | .10 |
| | ReLU | .52 | .96 | .97 | .98 | .98 | .10 |
| | ReLU Poly(3) | .13 | .94 | .86 | .98 | .69 | .10 |
| | Square | .27 | .10 | .40 | .16 | .10 | .10 |

Table 5: Test accuracy of the LeNet-5 model (LeCun et al., 1998) on MNIST for different optimization methods, learning rates, and activation functions. The optimization method used affects the performance of each activation function differently. The cosine activation offers a generic, robust construct that can be trained without the need of precisely choosing the learning rate.

Several important observations can be made from Table 5. When no activation function is used, the model achieves roughly the same performance for both optimization algorithms across different learning rates. On the other hand, the performance of other nonlinear activations changes with different optimization methods. In particular, for activation function that grow fast (i.e., $e^x - 1$, square), it is crucial to use an optimization method equipped with an adaptive learning rate like Adam to ensure stable training. By contrast, for a standard choice such as the linearly growing ReLU function, and a bounded function such as cosine, the model can learn with either optimizer without a precise tuning of the learning rate.

## B.2 TRAINING STABILITY AS A FUNCTION OF NETWORK DEPTH

In this section, we expand results presented in Figure 2 in the main text by including the performance on Higgs for every learning rate tried. Our conclusion remains the same as discussed in Section 6.2 which is that activation functions with bounded derivatives (e.g., cos, ReLU) can be trained with a wide range of learning rates, and different optimizers. By contrast, models relying on activation functions with high growth rate (e.g., exponential, square) can only be trained with a careful choice of the learning rate. This training instability is more pronounced when the number of layers is larger (deeper network).

| Optimizer | Learning rate
Activation | Number of layers | $1e^{-5}$ | $1e^{-4}$ | $1e^{-3}$ | 0.01 | 0.1 | 0.5 |
|---|---|---|---|---|---|---|---|---|
| SGD | Cosine | 0 | .62 | .63 | .64 | .64 | .64 | .64 |
| | | 1 | .57 | .65 | .68 | .69 | .69 | .58 |
| | | 2 | .55 | .65 | .69 | .70 | .70 | .56 |
| | | 3 | .54 | .63 | .70 | .71 | .71 | .57 |
| | | 4 | .52 | .60 | .70 | .69 | .71 | .57 |
| | ReLU | 0 | .62 | .63 | .64 | .64 | .64 | .64 |
| | | 1 | .58 | .66 | .68 | .69 | .69 | .61 |
| | | 2 | .56 | .66 | .70 | .70 | .70 | .63 |
| | | 3 | .55 | .65 | .70 | .70 | .70 | .56 |
| | | 4 | .54 | .64 | .70 | .71 | .71 | .56 |
| | Square | 0 | .62 | .63 | .64 | .64 | .64 | .64 |
| | | 1 | .59 | .65 | .66 | .66 | .66 | .55 |
| | | 2 | .56 | .65 | .68 | .56 | .50 | .50 |
| | | 3 | .54 | .61 | .52 | .50 | .50 | .50 |
| | $e^x - 1$ | 0 | .62 | .63 | .64 | .64 | .64 | .64 |
| | | 1 | .60 | .65 | .67 | .64 | .52 | .50 |
| | | 2 | .51 | .50 | .50 | .50 | .50 | .50 |
| | | 3 | .50 | .50 | .50 | .50 | .50 | .50 |
| Adam | Cosine | 0 | .62 | .64 | .64 | .64 | .64 | .63 |
| | | 1 | .65 | .68 | .69 | .69 | .65 | .50 |
| | | 2 | .66 | .70 | .71 | .71 | .60 | .50 |
| | | 3 | .66 | .71 | .72 | .72 | .60 | .50 |
| | | 4 | .66 | .71 | .72 | .60 | .60 | .50 |
| | ReLU | 0 | .63 | .64 | .64 | .64 | .64 | .63 |
| | | 1 | .65 | .68 | .69 | .70 | .67 | .50 |
| | | 2 | .66 | .69 | .71 | .71 | .69 | .50 |
| | | 3 | .66 | .71 | .72 | .72 | .64 | .50 |
| | | 4 | .66 | .70 | .72 | .72 | .50 | .50 |
| | Square | 0 | .63 | .64 | .64 | .64 | .64 | .63 |
| | | 1 | .64 | .66 | .67 | .67 | .67 | .66 |
| | | 2 | .65 | .68 | .69 | .69 | .69 | .69 |
| | | 3 | .57 | .68 | .70 | .70 | .64 | .59 |
| | $e^x - 1$ | 0 | .62 | .64 | .64 | .64 | .64 | .63 |
| | | 1 | .64 | .67 | .68 | .68 | .57 | .50 |
| | | 2 | .53 | .50 | .50 | .54 | .50 | .50 |
| | | 3 | .50 | .50 | .50 | .50 | .50 | .50 |

Table 6: Experimental results on Higgs. Vary the numbers of hidden layers.

## C  STRUCTURED WEIGHT MATRICES

Here we expand results in Section 6.3 and present more results showing the influence of structured weight matrices on the model performance. Table 7 and Table 8 give results in the same setting as in Table 2 presented in the main text, except that each dense layer is replaced with the proposed Hadamard layer (Table 7), and a dense layer with a low-rank weight matrix (Table 2).

| Activation | MNIST | Fashion-MNIST | Higgs | Criteo |
|---|---|---|---|---|
| Cosine | **98.5** | 88.9 | **70.3** | 58.0 |
| $e^x - 1$ | 94.0 | 85.9 | 50.0 | 57.4 |
| None | 92.2 | 83.6 | 63.5 | 48.5 |
| ReLU | 98.4 | **89.2** | 69.8 | 46.6 |
| ReLU Polyfit(3) | **98.5** | 88.7 | 62.4 | 52.7 |
| square | 45.4 | 35.7 | 54.4 | 25.6 |

Table 7: Test accuracy on all the datasets using the proposed Hadamard layer. We report test accuracy except Criteo where we report AUC-PR due to its skewed label distribution.

| Activation | MNIST | Fashion-MNIST | Higgs | Criteo |
|---|---|---|---|---|
| Cosine | 94.4 | 84.9 | **67.9** | 25.6 |
| $e^x - 1$ | 86.2 | 54.2 | 50.0 | 57.4 |
| None | 68.4 | 73.8 | 63.5 | 56.8 |
| ReLU | **94.8** | **86.3** | **67.9** | 57.7 |
| ReLU Polyfit(3) | 94.5 | 83.8 | 56.7 | 25.6 |
| Square | 60.6 | 35.0 | 50.0 | 25.6 |

Table 8: Test accuracy on all the datasets using low-rank dense layers. We report test accuracy except Criteo where we report AUC-PR due to its skewed label distribution.

For completeness, we also compare to other types of weight matrices: 1) PHD (Yang et al., 2015), 2) circulant matrix (Yu et al., 2018). The PHD approach is similar to our proposed Hadamard-Diagonal layer. The weight matrix is given by $W = PHD$ where $P$ is a sparse matrix with non-zero entries drawn from a Gaussian distribution, $H$ is the Hadamard matrix, and $D$ is the diagonal weight matrix. The PHD has the same number of parameters as our Hadamdard-Diagonal (HD) approach. The main difference is the presence of the sparse matrix $P$. The circulant matrix approach of Yu et al. (2018) also only requires $d$ weights parameters. The weight matrix is formed by forming a circulant matrix from these weights. We present a comparison of these three related approaches on Fashion-MNIST in Table 9.

We observe that our proposed Hadamard layer performs better than the PHD approach. This suggests that the presence of the sparse random matrix $P$ hurts the expressiveness of the model. It can be seen that the circulant weight matrix generally does not performance as well as the other two variants.

| Activation | Hadamard | PHD (Yang et al., 2015) | Circulant (Cheng et al., 2015) |
|---|---|---|---|
| Cosine | **88.6** | 88.1 | 80.3 |
| $e^x - 1$ | **80.2** | 59.7 | 10.0 |
| None | **83.7** | 83.5 | 83.6 |
| ReLU | 89.3 | 89.0 | **89.4** |
| ReLU Polyfit(3) | **88.5** | 87.6 | 87.8 |
| Square | 36.0 | 36.0 | **87.6** |

Table 9: Comparison of Hadamard transform against other structures imposed on the weight matrix. Notice how Hadamard works better across most activation functions.

