# OpenReview forum: "HD-cos Networks: Efficient Neural Architechtures for Secure Multi-Party Computation"
_ICLR.cc/2022/Conference — ICLR 2022 Submitted_

### Official Review · Reviewer_w59J · 2021-10-27

**Correctness:** 3
**Technical Novelty And Significance:** 2
**Empirical Novelty And Significance:** 2
**Recommendation:** 5
**Confidence:** 4

**Details Of Ethics Concerns:**

No obvious ethics concerns

**Main Review:**

Pros:
1. Compare to ReLU in MPC with multiple rounds of communication, proposed cosine function in 2PC setup only involves two rounds of communication. Since communication overhead is a big deal in MPC among parties, cosine approach decreases the cost.
2. Structure matrices used in the work can reduce bandwidth and save memory.
3. Theoretical explanation for both methods is provided to convince reader in some way.

Cons:
1. Since this work focuses on MPC approach, it is necessary to state what type of adversary model (semi-honest or malicious) is trying to deal with. More security analysis is expected if possible.
2. Although communication cost is analyzed in words, fail to provide table or charts to show efficiency compared with existing methods, since “efficient” is in the title. With tables or charts would convince more.

**Summary Of The Paper:**

MPC is a cryptographic technique to allow multiple party to jointly compute a protocol without leaking sensitive data, but building blocks in the neural network converted to MPC setup usually suffer from heavy communication overhead among parties, and jointly training ML models is also computationally expensive. So, in this work, authors propose an efficient MPC-based neural network. The network consists of cosine function as activation function under 2PC setup and linear transformations by Hadamard-Diagonal method.

**Summary Of The Review:**

To be honest, this paper is below acceptance standard because of innovation and lack of security analysis, although the writing of manuscript looks great. So, currently, I think I would not accept this paper and looking forward to see other reviewers’ comments to make final decision.

---

> ### Author Response · Authors · 2021-11-24
> **Initial response to Reviewer w59J**
>
> We thank the reviewer for taking the time to review our paper.
>
> > it is necessary to state what type of adversary model (semi-honest or malicious). More security analysis is expected.
>
> We assume that at least one of the two computation servers in our 2PC setup is honest. Most efficient MPC algorithms assume a semi-honest model. MPC algorithms designed for a malicious model tend to be more complex and have higher communication/computational cost. More importantly, as correctly noted by the reviewer, the goal of our work is to propose a neural network layer (HD-cos) that can be computed efficiently in an MPC setup. The computational benefits of our proposal remain regardless of the adversary model assumed.
>
> > Although communication cost is analyzed in words, fail to provide table or charts to show efficiency compared with existing methods
>
> We thank the reviewer for this suggestion. Computational efficiency is influenced by many factors. For example, using Intel AVX Intrinsics may speed up over equivalent C implementation for CPU intensive computation by a factor of 15x. Depending on the underlying implementation details, the exact runtime will vary. The scope of the present work is to confirm that the proposed HD-cos layer can give a neural network that performs as well as a ReLU based neural network with fully connected layers (which is widely considered to be a strong baseline), while having a manageable computational overhead. We show the latter by qualitatively evaluating computational costs by the number of communication rounds (see Table 1).

---

### Official Review · Reviewer_pKsC · 2021-11-02

**Correctness:** 1
**Technical Novelty And Significance:** 1
**Empirical Novelty And Significance:** Not applicable
**Recommendation:** 1
**Confidence:** 5

**Main Review:**

It seems to me that there is a fundamental flaw in how secret sharing is used: computing trigonometric functions on arithmetic secret shares makes no sense, as the shares are over a finite ring. I don't see a way to fix this problem.

**Summary Of The Paper:**

This paper proposes using a 2-party MPC protocol for evaluating neural networks on private data by replacing activation functions with a cosine function, and another technique of replacing fully connected layers with a specially structured matrix.

**Summary Of The Review:**

Unfortunately, it seems to me that this paper has a fundamental mistake and cannot be accepted.

---

> ### Author Response · Authors · 2021-11-24
> **Initial response to Reviewer pKsC**
>
> We thank the reviewer for taking the time to review our paper.
>
> > there is a fundamental flaw in how secret sharing is used: computing trigonometric functions on arithmetic secret shares makes no sense, as the shares are over a finite ring.
>
> We are unsure if we could follow the reviewer's reasoning. Could the reviewer please elaborate more on why it makes no sense?

---

> > ### Comment · Reviewer_pKsC · 2021-11-25
> > **Problem**
> >
> > Secret sharing operates in some finite ring. For example, operating in integers mod 13, 2 = 5 + 10 (mod 13), but cos(2) != cos(5 + 10).

---

### Official Review · Reviewer_g4Ma · 2021-11-02

**Correctness:** 2
**Technical Novelty And Significance:** 2
**Empirical Novelty And Significance:** 2
**Recommendation:** 3
**Confidence:** 3

**Main Review:**

While the ideas are interesting, the MPC aspects are insufficient:
- The paper does not name a secret sharing domain. Usually, numbers are shared modulo an integer in order to achieve security via uniform distribution of shares. This modular operation would hinder the summand-wise cosine and sine computations in Algorithm 1.
- It is common to use quantization with MPC because full floating-point computation is much more expensive. The authors don't specify how they have obtained their accuracy results. It would be insufficient to use floating-point computation because it is known that quantization reduces the accuracy [1].

The authors say that it is not efficient to implement ReLU accurately. However, there are implementations that do that, for example [2].

Minor:
- p4: (The) Kernel method is a type of powerful nonlinear
- p6: is a diagonal matrices (matrix)
- p6: Another example is (the) binary embedding

[1] https://arxiv.org/abs/1502.02551
[2] https://arxiv.org/abs/2104.10949



**Summary Of The Paper:**

The paper proposes changes to improve the performance of neural network training with multi-party computation, that is an activation function based on cosine and a linear functionality that reduces the number of weights to be trained.


**Summary Of The Review:**

The MPC aspects need more work.

---

> ### Author Response · Authors · 2021-11-24
> **Initial response to Reviewer g4Ma**
>
> We thank the reviewer for taking the time to review our paper. We indeed have in mind that ultimately, in a real MPC setup, numbers will be shared modulo an integer, as the reviewer correctly noted. However in all our experiments in the current work, we use floating-point computation to see the performance of the proposed HD-cos in the ideal setting i.e., little loss of numerical precision. Note that all other activation functions and other types of structured matrices (i.e., low rank) we compare with all use float-point numbers to provide a fair comparison. We believe that our results provide a reassurance that using cosine together with the Hadamard transform does not incur a loss of ML performance in practice. Investigating the ML performance of HD-cos under a lower numerical precision setting (e.g., ring of integers) will be an interesting topic for future research.
>
> > The authors say that it is not efficient to implement ReLU accurately. However, there are implementations that do that, for example [2].
>
> We thank the reviewer for sending these links. As far as we understand, [2] https://arxiv.org/abs/2104.10949 investigates the use of a GPU to accelerate computation of ReLU. The use of a GPU to accelerate computation is also a viable direction for HD-cos in future work.

---

> > ### Comment · Reviewer_g4Ma · 2021-11-26
> > **Insufficient**
> >
> > I think the insight has little bearing on actual secure computation because Algorithm 1 wouldn't work modulo an integer. Furthermore, [2] is only one example of exact ReLU computation, see https://eprint.iacr.org/2018/403 for an example without using GPUs.

---

### Official Review · Reviewer_7c3H · 2021-11-03

**Correctness:** 3
**Technical Novelty And Significance:** 3
**Empirical Novelty And Significance:** 3
**Recommendation:** 5
**Confidence:** 4

**Main Review:**

Strengths- The paper demonstrates the utility of cosine activation function and Hadamard transform in the privacy-preserving machine learning paradigm. The experimental results demonstrate the improvement on the prediction performance over benchmarks is promising.

Weaknesses-
1- The proposed approach combines well-known building blocks (e.g., cosine activation) that have been studied in the context of ML. Similarly, the secure computing component uses well-known cryptographic primitives such as Beaver's multiplication triplets.

2 - The proposed setup is for a (non-colluding) two-server model, where the two parties do not cooperate with each other. The two server model is restrictive in practical settings

3- The experiments do not include runtime comparisons between the baselines, or comparing the communication and computation costs, to demonstrate efficiency.

4- The baselines do not provide an extensive comparison with the former two-party MPC approaches (for instance SecureML [Mohassel-Zhang] also discussed in the related works section). Thus, it would be good to clarify the motivation on the choice of baselines.




**Summary Of The Paper:**

The paper proposes using a cosine activation function and Hadamard-Diagonal transformation as a means to improve the efficiency of MPC for machine learning. The paper considers a two-server model, where the training computation are carried out by two non-colluding parties. Experiments are provided to demonstrate the improvement on the test accuracy of the proposed approach over other baselines that use different activation functions.

**Summary Of The Review:**

Improving the efficiency and predictive performance of MPC-based techniques for machine learning is an important problem. It would be good to include experiments that demonstrate the efficiency over the benchmarks.

---

> ### Author Response · Authors · 2021-11-24
> **Initial response to Reviewer 7c3H**
>
> We thank the reviewer for taking the time to review our paper. In line with what the reviewer noted, the main goal of our work is to indeed experimentally demonstrate the utility of cosine activation and the Hadamard transform.
>
> > The proposed approach combines well-known building blocks (e.g., cosine activation) that have been studied in the context of ML.
>
> Our proposed combination of cosine as the activation function, and the Hadamard transform as a replacement for the fully connected layer is a result of careful analyses and intensive empirical studies. We assume that a feedforward neural network is to be deployed under a 2PC setup. Reducing the overall computational overhead amounts to reducing the computation at each layer. Our proposed cosine activation (in place of, say, the more popular choice of ReLU) is to **reduce the number of communication rounds** between the two MPC servers. Please see also pros of our approach listed by Reviewer w59J. The use of the Hadamard transform **reduces the number of parameters** from $d^2$ (where $d$ is the input dimension to a layer) to $d$. We think that judging the novelty of the two constructs by the fact that they are known (individually) is not a fair treatment to our proposal. We believe that it is the search for an appropriate combination through our qualitative as well as quantitative analyses that provides values. This is supported by our experimental results where we show that the combination gives neural networks that perform well in practice.
>
> > The proposed setup is for a (non-colluding) two-server model. Restrictive in practical settings.
>
> It is known that MPC can create a large computational overhead. This fact forms the basis for our proposal. We think that starting from the Two-Party Computation setup (as opposed to the more general Multi-Party Computation setup) will allow us to quickly make progress in this regard. Could the reviewer please help us understand why the 2PC setup is restrictive in practical settings?

---

> > ### Comment · Reviewer_7c3H · 2021-11-26
> > **Finite field**
> >
> > The assumption of 2PC is restrictive as in practical settings as they only allow collaboration between 2 parties and not generalize to the multi-party setup efficiently, but I now understand the authors' reasoning for using 2PC as a starting point. However, I also agree with the reviewer comments pointed out on the finite field implementation.

---

> > > ### Author Response · Authors · 2021-12-01
> > > **2PC**
> > >
> > > We thank the reviewer for further comment. To our knowledge, most MPC algorithms used in practice are 2PC. Could the reviewer please help send links to papers that discuss that MPC can be more practical?

---

### Decision · Program_Chairs · 2022-01-20

**Decision:**

Reject

**Comment:**

There appears to be to be a fundamental error in the paper, w.r.t. the application of the proposed approach to finite fields. As a result, the paper cannot be accepted in its current form.